## Original Research Article

mathematical model; plant biomechanics; proprioception; quantitative plant biology; shoot gravitropism.

**Author for correspondence:**
Satoru Tsugawa,
E-mail: stsugawa@bs.naist.jp
Satoru Tsugawa and Tomohiko G. Sano contributed equally to this study.

# A mathematical model explores the contributions of bending and stretching forces to shoot gravitropism in Arabidopsis

Satoru Tsugawa[1] [iD], Tomohiko G. Sano[2] [iD], Hiroyuki Shima[3] [iD], Miyo Terao Morita[4] [iD], and Taku Demura[1] [iD]

[1]Graduate School of Science and Technology, Nara Institute of Science and Technology, Ikoma, Japan; [2]Flexible Structures Laboratory, Institute of Mechanical Engineering, EPFL, Lausanne, Switzerland; [3]Department of Environmental Sciences, University of Yamanashi, Kofu, Japan; [4]Division of Plant Environmental Responses, National Institute for Basic Biology, Okazaki, Japan

## Abstract

Plant shoot gravitropism is a complex phenomenon resulting from gravity sensing, curvature sensing (proprioception), the ability to uphold self-weight and growth. Although recent data analysis and modelling have revealed the detailed morphology of shoot bending, the relative contribution of bending force (derived from the gravi-proprioceptive response) and stretching force (derived from shoot axial growth) behind gravitropism remains poorly understood. To address this gap, we combined morphological data with a theoretical model to analyze shoot bending in wild-type and *lazy1-like 1* mutant *Arabidopsis thaliana*. Using data from actual bending events, we searched for model parameters that minimized discrepancies between the data and mathematical model. The resulting model suggests that both the bending force and the stretching force differ significantly between the wild type and mutant. We discuss the implications of the mechanical forces associated with differential cell growth and present a plausible mechanical explanation of shoot gravitropism.

## 1. Introduction

Plants exhibit various tropisms, processes in which their body bends according to directional environmental cues. For example, in gravitropism, shoots grow upward (negative gravitropism) to promote efficient photosynthesis and reproduction (Blancaflor & Masson, 2003; Firn & Digby, 1980; Haswell, 2003; Morita & Tasaka, 2004; Perbal & Driss-Ecole, 2003) and roots grow downward (positive gravitropism) to absorb water and nutrients effectively (Blancaflor & Masson, 2003; Firn & Digby, 1980; Morita & Tasaka, 2004; Perbal & Driss-Ecole, 2003). When shoots or roots bend, the cells within the organ sense their positions and orientations by specific physiological or mechanical means (Haswell, 2003; Morita & Tasaka, 2004; Perbal & Driss-Ecole, 2003) and then induce differential growth between the inner and outer flanks of the organ (Blancaflor & Masson, 2003; Firn & Digby, 1980). In addition to this gravi-sensing mechanism to induce bending, plants have a mechanism to straighten shoots when they bend too much, indicating that the amount of bending depends not only on the inclination angle of the organ but also on its curvature (Bastien *et al.*, 2013; Okamoto *et al.*, 2015).

Several studies have examined the molecular mechanism of directional sensing in gravitropism (Blancaflor *et al.*, 1998; Casper & Pickard, 1989; Friml, 2003; Fukaki *et al.*, 1998; Kiss *et al.*, 1989; Moulia *et al.*, 2006; Moulia & Fournier, 2009; Tasaka *et al.*, 1999; Tsugeki *et al.*, 1998). In the initial step, amyloplasts in specific cells (e.g., the columella cells in roots) are thought to travel downward owing to gravity, a process called sedimentation of amyloplasts (Blancaflor *et al.*, 1998; Fukaki *et al.*, 1998; Morita & Tasaka, 2004). This sedimentation subsequently turns on a specific physical or chemical switch that induces auxin transport in the direction of gravity, although the detailed mechanism of the switch is unknown (Moulia *et al.*, 2006; Moulia & Fournier, 2009). In the second step, the different concentrations of auxin in the inner and outer flanks of shoots/roots cause differential cell growth (DCG) and hence bending (Friml, 2003). Various types of mutants that affect these processes have been isolated (Casper & Pickard, 1989; Fukaki *et al.*, 1998; Kiss *et al.*, 1989; Taniguchi *et al.*, 2017; Tasaka *et al.*, 1999). For example, the *Arabidopsis thaliana lazy1-like 1*(*lzy1*) mutant exhibits reduced gravitropism and slow bending in shoots. In this mutant, amyloplast sedimentation is normal but the early steps of gravity signal transduction to switch the DCG are disrupted (Taniguchi *et al.*, 2017). To understand how the molecular mechanisms of gravitropism relate to organ bending, researchers have recently turned to data- and model-approaches (Bastien *et al.*, 2014; 2016; Basu *et al.*, 2007; Chelakkot & Mahadevan, 2017; Coutand *et al.*, 2007; Kutschera, 2001; Moulia *et al.*, 2019; Philippar *et al.*, 1999).

Pioneering data studies of gravitropism were based on the development of image analysis tools to quantify the tip angles of shoots and roots (Coutand *et al.*, 2007; Kutschera, 2001; Philippar *et al.*, 1999). However, such a simple angle extraction often fails to account for the actual morphology of bending organs (especially for shoots) because the shoot has a continuous filament-like structure and bending involves changes in curvature and length that are distributed along the entire shoot [the limitations of quantifying only tip angles are summarized in Moulia and Fournier (2009). To solve this problem, a new technique that quantifies continuous displacement by adding a distinctive randomized pattern to an object was developed to track the growth of roots [KineRoot, Basu *et al.* (2007)]. Recently, KymoRod was developed to recognize root angle, curvature and relative growth rate without patterns (Bastien *et al.*, 2016). The data approach is now widely applicable with high precision but omits mechanical information, such as stretching and bending force, that can only be extracted using model approaches.

Researchers have constructed a variety of mathematical models to describe the shoot's morphology and bending during gravitropism (Bastien *et al.*, 2013; 2014). Early models were based on active processes where the curvature changes depending on the horizontal angle of the stem segment relative to the ground (gravity sensing) and the curvature of the stem segment (straightening or proprioception) (Bastien *et al.*, 2014). Although some of these models have analytical solutions, analytical studies are not applicable to all types of shapes after bending. Recently, based on the active process mentioned above (Bastien *et al.*, 2014), a numerical method that ensures the minimization of elastic energy has been developed including the effects of elasticity; this method explains how stationary shapes are affected by the inclination angle and curvature of the shoots (Chelakkot & Mahadevan, 2017; Moulia *et al.*, 2019). One of these studies (Chelakkot & Mahadevan, 2017) revisits the concept of 'morphospace', a well-known methodology in evolutionary and anatomical studies (Eberle *et al.*, 2014; Polly & Motz, 2016) whereby representing a possible form, shape and structure of an organism only by a few characteristic parameters (e.g., width, height, growth rate, and so forth). This type of modelling approach is important because it incorporates the mechanical aspects of shoot gravitropism.

Despite these advances in data and model approaches, the role of mechanical forces during bending of plant organs is poorly understood. The purpose of this study was to elucidate the mechanics of gravitropism using mathematical model based on actual data. First, we analyzed the morphological differences between wild-type Arabidopsis and the *lzy1* mutant during shoot bending. Next, we combined the data with the previously reported model in Chelakkot and Mahadevan (2017) to extract mechanical forces applied to the shoot during gravitropism. Finally, we formulated the biological implications of these mechanical forces and discussed possible experiments to verify them.

## 2. Results

### 2.1. Maximum curvature after 140 min of bending is significantly lower for the lzy1 mutant than for the wild type

We quantified the morphological state of shoots before and during bending, by reanalyzing gravitropic responses which had been reported previously (Taniguchi *et al.*, 2017). Time-lapse images of wild type and *lzy1* mutant inflorescence stems bending in response to gravity are shown in Figure 1a,b. We used ImageJ to extract the centerline of each shoot from the original images and constructed a

continuous description of the centerline by spline interpolation (see Section 4). Based on the continuous curve, the evolution of shoot lengths and curvatures were calculated for 16 wild-type and for 15 *lzy1* mutants. Three representatives of each genotype are shown in Figure 1c,d (see all examples in Figure SI2). We set the origin of the curvilinear abscissa (mm) at the base of the shoot, with values increasing with distance along the centerline from the base to the tip. The curvature at the middle of the wild-type shoot changed to positive values (red) after 40–50 min (Figure 1c) but was lower for *lzy1* mutant shoots (light-red) (Figure 1d).

For a morphological description, we produced a morphospace (Figure 1e) using the extension ratio of shoot length (corresponding to shoot strain) and the maximum curvature at 140 min as characteristics of shoot shape. The extension ratio was calculated by comparing the initial and final shoot lengths $\left(L_{t_e} - L_{t_s}\right)/L_{t_s}$, where $L_{t_s}$ ($L_{t_e}$) denotes the shoot length at the starting (ending) time $t_s$ ($t_e$) after placing the shoot in an almost horizontal position. We note that the morphospace can be described using other characteristics (e.g., inclination angle, timing of bending, and so forth), but we chose the two characteristics above for the following reasons. The extension ratio enables us to check the consistency of our results with the reported observation that the *lzy1 lzy2 lzy3* triple mutant does not show significant growth impairment (Taniguchi *et al.*, 2017, which we assumed would also be the case for the *lzy1* mutant. Using the maximum curvature, it is possible to check if there is a significant difference in the degree of bending between the wild type and the *lzy1* mutant, as reported in Taniguchi *et al.* (2017). Consistent with these previous results, the morphospace indicates that the extension ratio of the *lzy1* mutant is not significantly different from that of the wild type, but the maximum curvature at 140 min is significantly lower in the mutant than in the wild type (Figure 1f).

### 2.2. A mathematical model captures typical shoot bending events in the wild type and the lzy1 mutant

To understand the mechanics underlying the different morphologies of the wild type and the *lzy1* mutant, we reimplemented the mathematical model proposed in Ref. Bastien *et al.* (2013, 2014), Chelakkot and Mahadevan (2017) (see Section 4). We note that the mathematical model itself in this study is exactly same as the model in Chelakkot and Mahadevan (2017) but the fitting of the experimentally obtained bending dynamics data to this model is novel in the context of gravitropism. In this model, the plant shoot is modeled as a growing elastic rod under gravity. The rod is separated into growing and nongrowing zones. In the growing zone, the intrinsic (or spontaneous) curvature $\kappa^*$ and the natural length of the stem segments change as functions of time during gravitropic and proprioceptive responses (Moulia *et al.*, 2019). At each time step, the shoot configuration is determined through force and momentum balance equations of the rod under the target geometry. By contrast, in the nongrowing zone, we do not change the target geometry. To compute the shoot shape of the model, we discretized the shoot centerline into a set of particles (vertices) connected by elastic springs (Figure 2a left). The positions of the particles are determined by the balance of stretching, bending and gravitational forces (Figure 2a right). We repeated the updates of the target geometry and the current configuration to compute the shoot shapes.

This model system has two characteristic dimensionless parameters: the growth-sensitivity parameter $S$ and growth-elasto-gravity parameter $\varepsilon$. The parameter $S$ represents the ratio of the size of

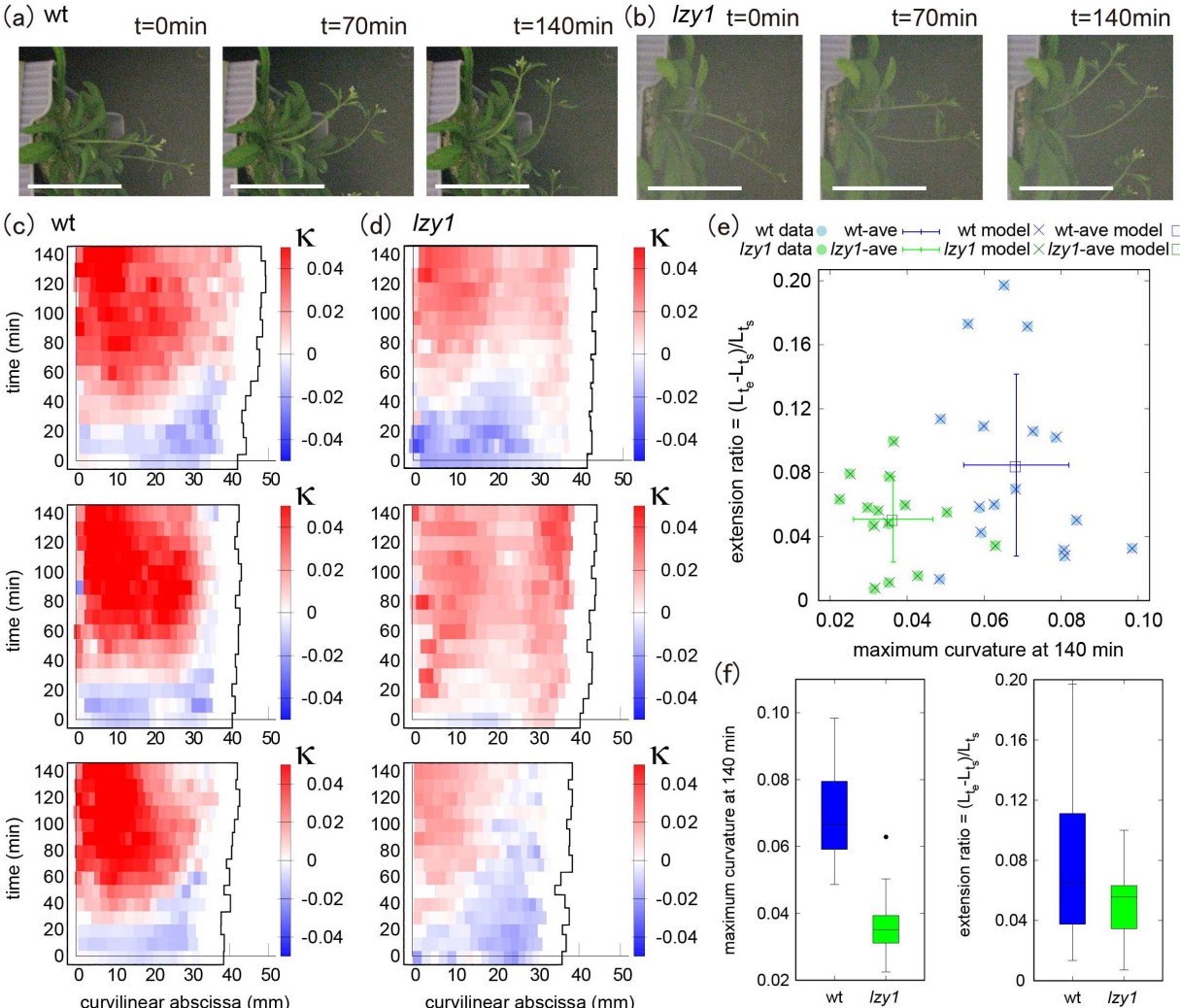

**Fig. 1** (a, b) Time-lapse images of shoot gravitropism for the wild type ($t$ = 0, 70 and 140 min) (a) and *lzy1* mutant (b), with scale bars representing 50 mm. (c, d) Colour diagram of the curvature of the shoot as a function of time (min) and curvilinear abscissa (mm) of three samples for the wild type (c) and of three samples for the *lzy1* mutant (d). (e) Morphospace diagram with maximum curvature and extension ratio for data and model for the wild type (blue) and for the *lzy1* mutant (green). The square plots represent the averaged values for the wild type (blue) and for the *lzy1* mutant (green). Error bars are standard deviations. (f) Boxplot of the maximum curvature at 140 (min) showing a statistically significant difference in maximum curvature between the wild type and *lzy1* mutant with $t$-test $p < .005$ (left panel), and not in the extension ratio (right panel).

the growing zone $l_g$ relative to the sensitivity length $l_s$, that is, $S = l_g/l_s$. The parameter $\varepsilon$ is the ratio of the elasto-gravity length $l_e = (B/\rho g)^{1/3}$ and the size of the growing zone, that is, $\varepsilon = l_g/l_e$ (see also Section 4).

Based on the morphospace in Figure 1e, we systematically searched through values for the dimensionless parameters $S$ and $\varepsilon$ to find those that resulted in the minimum deviation between the model and data. We note that the search was done with fixed $l_g$ with the experimentally determined initial length, and with various $l_e$ for $\varepsilon$ by changing mass density per unit length, $\rho$ and with various $l_s$ for $S$ by changing both gravi-sensitivity $\beta$ and the proprioceptive sensitivity $\gamma$ (see Section 4). To extract the typical bending behaviour of the wild type and the *lzy1* mutant, we identified the best-fitted parameters with the averaged values for the wild type (wt-ave model) and *lzy1* mutant (*lzy1*-ave model). The extracted values of gravitropic sensitivity $\beta$ in wild type ($\beta = 2.14 \pm 1.95$) was higher than those in *lzy1* mutant ($\beta = 1.38 \pm 1.30$), reflecting the defects in DCG in *lzy1* mutant. On the other hand, the extracted values of proprioceptive sensitivity $\gamma$ in wild type ($\gamma = 0.08 \pm 0.07$)

was as same level as those in *lzy1* mutant ($\beta = 0.11 \pm 0.08$), indicating the same level of proprioceptive strength.

The averaged models exhibit similar bending behaviours as the real experimental data, where the wild type bends to a greater extent than the mutant does, as shown in Figure 2c (wild type) and Figure 2d (*lzy1* mutant). While the model does not reproduce the shapes of the shoot perfectly due to the variation of the initial conditions in data (see Figure SI5), the spatio-temporal maps of the curvature were not significantly different from the data. The model predicts a spatio-temporal trend in curvature increasing from the tip to the bottom of the shoot overtime, and those decreasing at tip after 100 min (Figure 2e,f), which are consistent with the real curvature data (Figure 1c,d).

## 2.3. The shoot models provide information about the mechanical forces involved in bending

The configuration of the model shoot is realized by the balance between the forces derived from the biological active process (the stretching and bending forces), andexternal gravitational force.

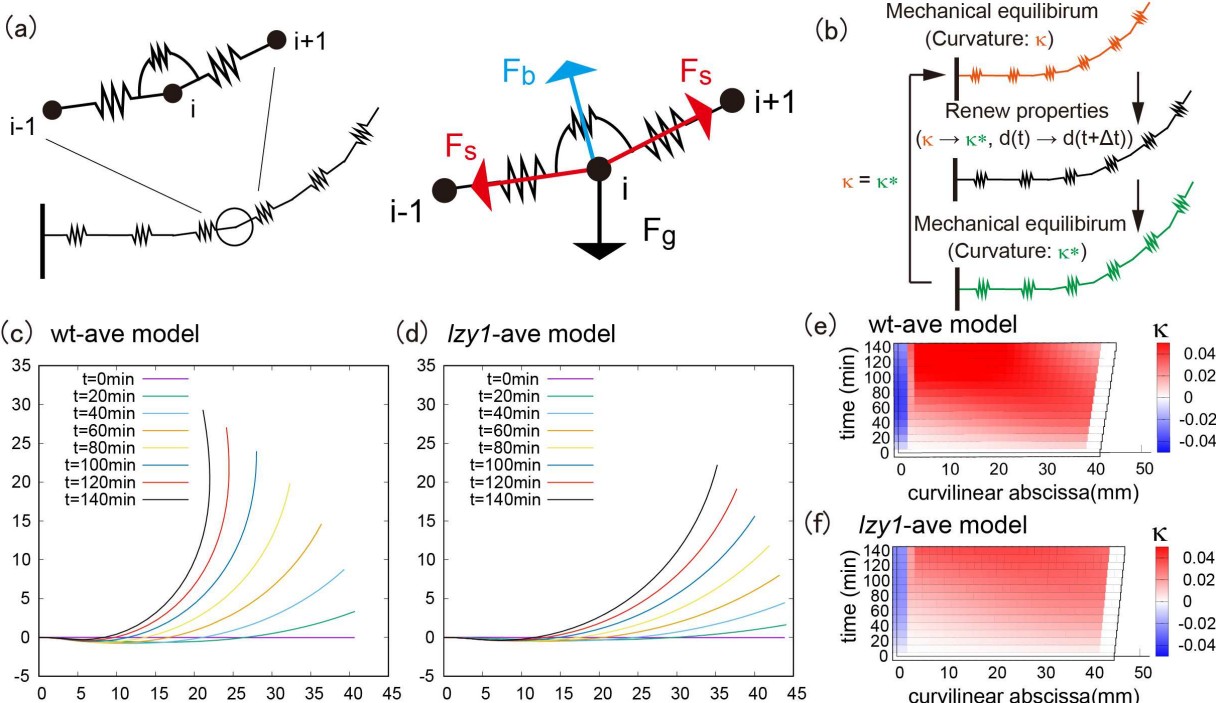

**Fig. 2** (a) Schematic illustration of the mathematical model and all applied force at the vertex. (b) Flowchart of the simulation loop; after the mechanical equilibrium at each step, the intrinsic curvature and length between neighboring vertices are updated, and another mechanical equilibrium will be satisfied. This loop is repeated until the shoot reaches the desired morphological shape. (c–d) Typical bending morphology in the wt-ave model (c) and the *lzy1*-ave mutant (d). (e–f) Colour diagram of the curvature in the wt-ave model (e) and *lzy1*-ave mutant (f).

We note that our model shoot is inextensible and unshearable filament as the model in Chelakkot and Mahadevan (2017) or other models of growing elastic rod in Goriely (2017) at every mechanical equilibrium configuration. The stretching and bending forces we discuss in the next section originate from changes in natural length and intrinsic curvature, respectively. To clarify the role of the stretching and bending forces, we decomposed the total displacement of the shoot $\vec{u}$ at each time step into the individual displacements from the stretching force $\vec{u}_s$, the bending force $\vec{u}_b$ and gravity $\vec{u}_g$ in the wt-ave model (Figure 3a,b) and in the *lzy1*-ave model (Figure 3c,d). We note that the displacement from gravity was negligible both in wt-ave and *lzy1*-ave model.

In both the wt-ave and the *lzy1*-ave models, the shoot tip deforms greatly, but the displacement at the base is small (Figure 3a1–d1). We noticed that the stretching and bending forces work differently at the tip and at the base. The direction of the stretching force near the tip is opposite to the bending direction, implying that the stretching force near the tip suppresses bending (a proprioceptive effect), as shown by the blue-filled arrow in Figure 3b2. However, bending near the tip is promoted by the direction of the bending force, as shown by the red-filled arrows in Figure 3b3. Near the base, by contrast, the stretching force promotes bending, whereas the bending force suppresses bending (red and blue open arrows in Figure 3b2,b3, respectively). Because there is little morphological change at the base, it is expected that the effects of these two forces on bending are comparable in magnitude, resulting in no significant change at the base.

The *lzy1*-ave model exhibits spatio-temporally similar behaviour, although the colour diagram shows less displacement and smaller changes in stretching and bending forces compared to the wt-ave model (Figure 3c,d). Accordingly, the maximum stretching force and maximum bending force are significantly weaker in the *lzy1*-

ave model compared to those in the wt-ave model (Figure 3e,g). Quantitatively, the magnitudes of these forces are about 50–60% smaller than those in the wt-ave model. That means that the defects in the signal transduction from the gravi-sensing in *lzy1* mutant are reflected in the weaker magnitudes of stretching and bending force.

## 2.4. Biological implications for gravitropism, proprioception and shoot axial growth

We have clarified that the bending and stretching forces have different effects on shoot morphology during bending. The remaining question is whether the decomposed bending and stretching forces can be translated into biological counterparts. The biological process underlying the change of the curvature is differential cell growth (DCG), which is realized by a combination of the gravi-proprioceptive response and the shoot axial growth as discussed below.

The bending force controls the degree of gravity-free bending and is associated with both gravitropism and proprioception through the change in the intrinsic curvature $\kappa^*$. Gravitropism promotes bending, whereas proprioception suppresses bending, as illustrated in Figure 4a. The former and latter depend on the inclination angle and the curvature, respectively. We call this combined effect gravi-proprioceptive response (GPR). The gravitropic response becomes significant when a stem segment has a low inclination and low curvature because, in this case, gravitropism is stronger than proprioception, resulting in the promotion of bending (Figure 4a top). By contrast, a high inclination and high curvature suppress bending because proprioception is dominant (Figure 4a bottom). Therefore, the bending forces for the wild type are significant at the start of the bending process and become smaller, starting from the tip, as the shoot bends (Figure 3a3,c3).

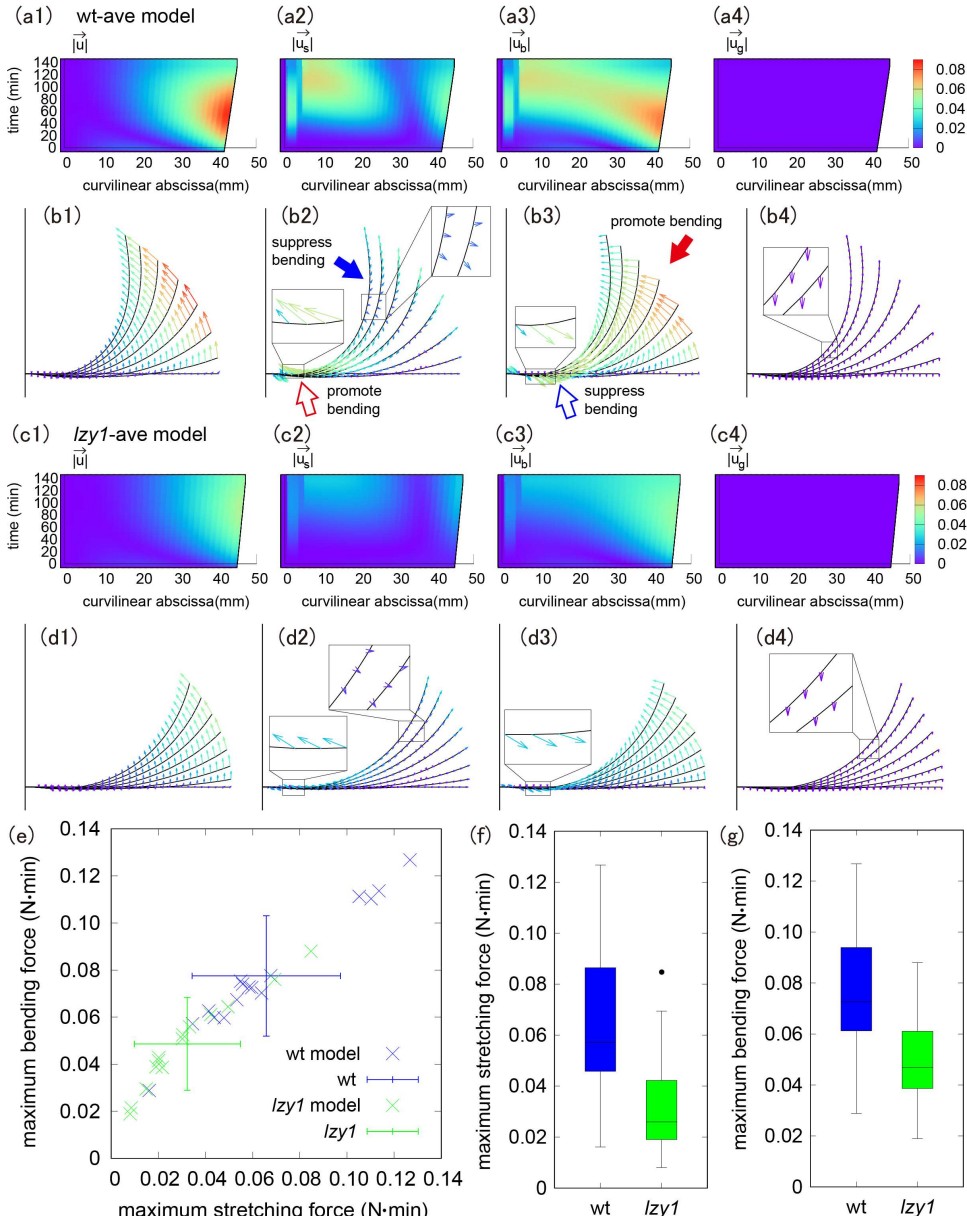

**Fig. 3** (a b) Colour diagram of magnitude and direction of the displacement of the shoot (a1, b1), stretching force (a2, b2), bending force (a3, b3) and gravity (a4, b4) in the wt-ave model. (c, d) Colour diagram of magnitude and direction of the displacement of the shoot (c1, d1), stretching force (c2, d2), bending force (c3, d3) and gravity (c4, d4) in *lzy1*-ave model. (e) Scatter plot of the maximum stretching and bending using the model. (f,g) Boxplot of the maximum stretching force (f) and bending force (g), showing a statistically significant difference with *t*-test $p < .005$. The dot represents outliers excluded in the statistical test.

The stretching force is determined by shoot axial growth and is governed by both the growth and the curvature of the stem segments. As the shoot grows, it changes the intrinsic curvature $\kappa^*$ and the current curvature $\kappa$ is governed by mechanical balance. We call this combined effect growth-curvature-induced bending (GCB), which is suppressed as the shoot grows because the corresponding tensional forces act in the direction opposite to the bending direction, thereby suppressing bending (Figure 4b top). By contrast, GCB is enhanced in the absence of growth because the contracting forces act in the same direction as bending (Figure 4b bottom). For the wild type, therefore, the stretching force is less significant at the start of bending, and is reinforced around the base, as shown in Figure 3a2,c2.

As a consequence of the two bending effects, the dynamic shoot morphology (i.e., curvature) is determined. Assuming the shoot is a solid material with thickness $2\delta$ and relative elongation growth rates $\dot{\varepsilon}_{\text{outer}}$ and $\dot{\varepsilon}_{\text{inner}}$ for the outer flank and inner flank, respectively (Figure 4c), the degree of the DCG is defined as $\Delta(s,t) = (\dot{\varepsilon}_{\text{inner}} - \dot{\varepsilon}_{\text{outer}})/(\dot{\varepsilon}_{\text{inner}} + \dot{\varepsilon}_{\text{outer}})$ [see Ref. Bastien *et al.* (2014)]. We then quantified the DCG as $\dot{L}_0 \Delta(s,t) = \dot{\varepsilon}_{\text{inner}} - \dot{\varepsilon}_{\text{outer}}$. DCG is negative if the outer cells grow faster than the inner ones; otherwise DCG is positive. The DCGs for both the wild type and *lzy1* mutant are negative when the shoot bends upward, but after 100 min, when the shoot straightens, DCG becomes positive near the tip (Figure 4d). These behaviours of the DCGs are consistent with the fact that displacement becomes large at the tip early in the bending process, then diminishes after 100 min (Figure 3a1,c1). Therefore, the mechanics of the stretching and bending force discussed above could be interpreted as the GPR and GCB, respectively, and the consequence of the two effects can be observed in DCG.

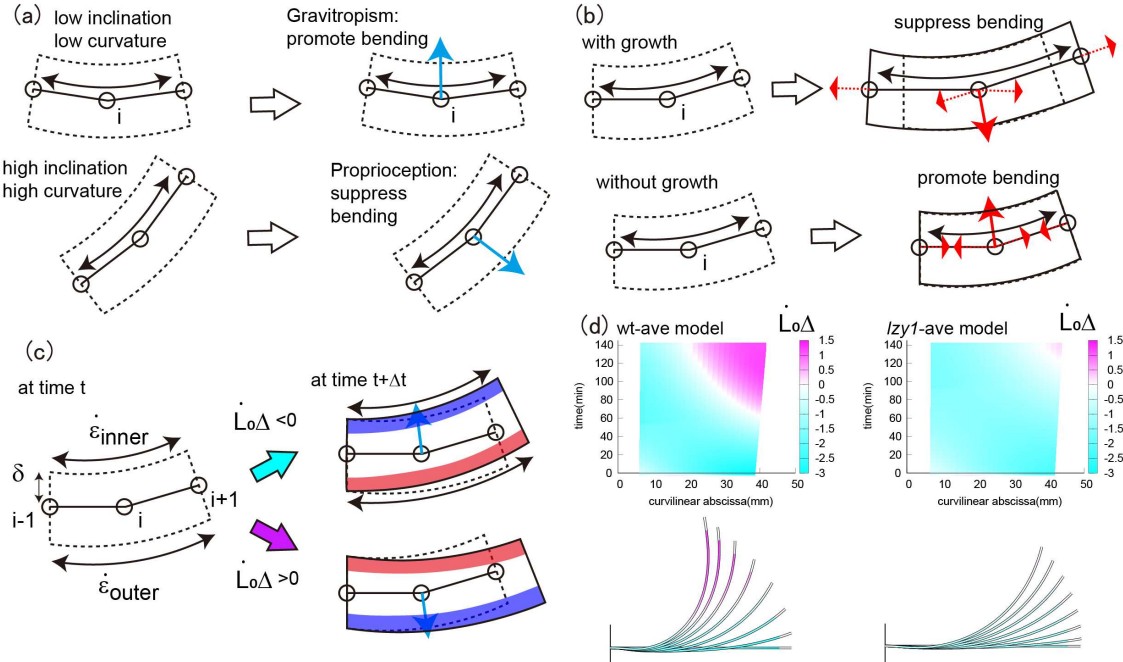

**Fig. 4** (a) Schematic illustration of the GPR. The GPR is strong if the stem segments have a low inclination and low curvature (top) and weak if they have a high inclination and high curvature (bottom). (b) Schematic illustration of the GCB. The GCB is low when there is shoot growth (top) and is high in the absence of shoot growth (bottom). (c) Schematic illustration of DCG. (d) Colour plot of DCG for the wild type (left) and *lzy1* mutant (right).

### 2.5. Future perspective: prediction for growth at the cellular level

The mathematical model provides a quantitative assessment of the differential growth at the cellular level. At the initial bending stage, the relative elongation growth rate of the outer flank at tip in the wild type was $\dot{\varepsilon}_{\text{outer}} = 2 \times 10^{-4}\ \left(\text{min}^{-1}\right)$ and that of the inner one $\dot{\varepsilon}_{\text{inner}} = -2 \times 10^{-4}\left(\text{min}^{-1}\right)$ (see details for $\dot{\varepsilon}_{\text{inner}}$ and $\dot{\varepsilon}_{\text{outer}}$ in Figure SI6), implying that the inner flank shrinks as the shoot grows. This counter-intuitive prediction at the cellular level may be experimentally clarified by measuring the growth rates of cells at the inner and outer franks for wild type and for several mutants with defect in building secondary cell wall (Kubo *et al.*, 2005; Mitsuda *et al.*, 2008) and with defect in posture control of shoot (Okamoto *et al.*, 2015).

### 3. Discussion

In this study, we extracted the detailed morphologies of wild type and *lzy1* mutant Arabidopsis shoots during bending. We confirmed that their maximum curvatures were significantly different, whereas their extension ratios were not. Subsequently, we fitted our mathematical model to the bending events and searched the best fitted dimensionless parameters, $S$ and $\varepsilon$. We then constructed mathematical models corresponding to the averaged bending behaviours of the wild type and the *lzy1* mutant and confirmed that the reconstructed bending events in the models were similar to the actual bending events in the data. With this setup, we could explore the mechanical information associated with stretching, bending and gravity, which is not observable in experiments.

The mechanics underlying the shoot bending events can be summarized as follows. The biological process underlying the change of the curvature is DCG, which can be interpreted as a combined result of the GPR and the GCB, which correspond to the bending force and the stretching force, respectively. The bending force (GPR) becomes larger if gravitropism is more significant than proprioception. Shoot growth stretches the shoot axially and induces another bending deformation, called GCB. As a consequence of the two bending effects, we can understand the dynamic shoot morphology in terms of mechanics. These mechanical aspects of bending are the underlying mechanism hidden in the experimental description of shoot gravitropism.

For the improvement of the mathematical model, more realistic growth or bending models will need to be developed (e.g., the effect of lignification of secondary cell walls (Chelakkot & Mahadevan, 2017) and the temporal delay in gravitropic response (Agostinelli *et al.*, 2020; Chauvet *et al.*, 2019). For the improvement of fitting between data and model, in addition to our restricted two parameters (maximum curvature after 140 minutes and extension ratio), the spatio-temporal curvature will need to be made correspondence for further understanding. This type of interdisciplinary study highlights the potential of connecting experimental observations with theoretical models and leads to a richer understanding of shoot gravitropism.

### 4. Materials and methods

#### 4.1. Data of gravitropism

We reanalyzed the data of gravitropic responses of wild type and *lzy1* mutant which had been reported in our previous study (Taniguchi *et al.*, 2017). We used *A. thaliana* Columbia-0 as the wild type and *lzy1* (GABI_591A12). Surface-sterilized seeds were sown on MS plates [1 × Murashige Skoog salts, 1% (w/v) sucrose, 0.01% (w/v) myoinositol, 0.05% (w/v) MES (2-(N-morpholino) ethanesulfonic acid) and 0.5% (w/v) gellan gum; pH 5.8], incubated in darkness at 4°C for 2–3 days, grown at 23°C in a growth chamber under continuous light for 10–14 days, transplanted

to soil, and grown under continuous light. Plants with primary inflorescence stems of 4–8 cm in length were gravistimulated by placing horizontally under nondirectional dim light at 23°C after 1 hr of preincubation. Photographs were taken at indicated times. 15 or 16 individuals of wild type or *lzy1* mutant were tested, respectively.

## 4.2. Extraction of angle and curvature of shoot in data

To extract continuous angle information of the shoot, we used ImageJ to detect the positions of the shoot at intervals of approximately 0.2 mm. Subsequently, we interpolated the detected points (approximately 20 points) by a third-degree spline interpolation (Figure SI1).

## 4.3. Mathematical model

We review the mathematical model proposed in Chelakkot and Mahadevan (2017), where gravitropic kinematics and mechanics are combined. The centerline of the shoot is modelled by a curve, whose position vector at time $t$ is given by $\boldsymbol{r}(s,t) = (x(s,t), y(s,t))$. Here, $s$ represents the arc-length of the shoot, satisfying $0 \leq s \leq L(t)$, with the total length $L(t)$ at time $t$. We introduce the bending angle $\theta(s,t)$, as the angle between the local tangent and $y$-axis. Here, the tangent vector of the shoot is described as $\hat{t} = (\sin\theta, \cos\theta)$. We clamp the basal end ($s = 0$) as $\theta(0,t) = -\pi/2, x(0,t) = y(0,t) = 0$, and the apical end ($s = L$) is set to be free for the force and moment at any time $t$. The shoot grows uniformly at speed $\dot{L}_0$ in the growing zone of length $\ell_g$ from the apical end. In the growing zone, the local intrinsic curvature of the shoot $\kappa^*(s,t)$ is modified by the active bending of the shoot, such as the local gravitropism and the proprioception for the current shoot configuration $\theta(s,t)$. The change of $\kappa^*$ in time is described as

$$\frac{1}{\dot{L}_0}\frac{\partial \kappa^*}{\partial t} = -\beta\sin\theta - \gamma\kappa, \tag{1}$$

where $\beta$ and $\gamma$ are gravitropic and proprioceptive sensitivities, respectively (Bastien *et al.*, 2014). The shape of the model shoot is determined by the moment and force balance of the rod with the intrinsic (inhomogeneous) curvature at every time $t$ (i.e., the speed of the growth is assumed to be sufficiently slower than that of the mechanical relaxation). $M(s,t)$ and $\boldsymbol{F}(s,t) = (H(s,t), V(s,t))$ are, respectively, the internal moment (around $z$-axis) and force acting on the cross section of the shoot at the arc-length $s$ and time $t$. The moment and force balance equations are given by

$$\frac{\partial M}{\partial s} + V\sin\theta - H\cos\theta = 0, \quad \frac{\partial H}{\partial s} = 0, \frac{\partial V}{\partial s} = -\rho g, \tag{2}$$

with the constitutive law given by $M = B(\kappa - \kappa^*)$. We have introduced the bending modulus $B$, the mass density per unit length $\rho$ and the gravitational acceleration $g$.

Three different length scales exist in this model system: the characteristic sensitivity length $\ell_s = \gamma/\beta$, the length of the growing zone $\ell_g$ and the elasto-gravity length (persistent length of elastic bending against gravity) $\ell_e = (B/\rho g)^{1/3}$. Thus, the two critical dimensionless parameters are the growth-sensitivity parameter $S = \ell_g/\ell_s$ and the growth-elasto-gravity parameter $\varepsilon = \ell_g/\ell_e$.

To simulate the shoot morphology, we discretized the centerline into a set of connected particles as $\boldsymbol{r}(s,t) \to \boldsymbol{r}_i(t)\,(i = 1, 2, \ldots, N)$, where $i$ represents the index of the node (vertex). The position of the node $\boldsymbol{r}_i(s,t)$ is updated based on the force and moment balance

equation (2). The elastic force at the $i$th node is computed from the stretching of the bond $b_i = |\boldsymbol{r}_{i+1} - \boldsymbol{r}_i|$ and the angle of adjacent bonds $\phi_i$. The stretching and bending potentials are respectively given by $U_s = \left(\frac{E}{2}\right)\sum_i(b_i - d_i)^2$ and $U_b = \left(\frac{B}{2}\right)\sum_i\left(\phi_i - \phi_i^*\right)^2$. The indices $d_i$ and $\phi_i^*$ are the natural length of the bond and the natural angle of adjacent bonds (calculated from $\kappa^*$), respectively, which will be updated by the growth rule and equation (1). As assumed in Chelakkot and Mahadevan (2017), we chose the rod elasticity parameters such that the rod is practically inextensible. The typical dimensions of the shoot in our experiments are the initial shoot length $L_0 = 40$ mm (natural length of the bond $d \sim 1.33$ mm) and the radius $\delta = 0.5$ mm. The shoot has the Young's modulus of $E_y \sim 10$ MPa and the bending modulus $= E_y\left(\pi\delta^4/4\right) \sim 5 \times 10^{-7}$ Nm$^2$. The typical stretching force $f_s^*$ and bending force $f_b^*$ of an elastic rod that has the same $E_y$ and D are estimated as $f_s^* = E_y\pi\delta^2 \sim 8$N and $f_b^* = D/L_0^2 \sim 3 \times 10^{-4}$N, respectively, from which we find $f_s^* \gg f_b^*$. In the discretized simulation, this condition is realized by setting $EL_0^2/B \gg 1$. We adapted the elasticity parameters $E$ and $B$ computed from those of a shoot via $E = E_y\pi\delta^2/d$ and $B = D/d$, which satisfies $f_s^* \gg f_b^*$. In this paper, as the extension ratio of wild type and *lzy1* mutant were similar (see Figure 1), we assumed that the elastic properties $E$ and $B$ for wild type are the same as those for *lzy1* mutant. The total force applied on the vertex $i$ is then given by

$$\boldsymbol{F}_i = -\nabla_{r_i}U_s - \nabla_{r_i}U_b - \nabla_{r_i}U_g, \tag{3}$$

where $U_g$ represents the gravitational potential. In the main text, we assign the stretching, bending and gravitational forces as $\boldsymbol{F}_s = -\nabla_{r_i}U_s$, $\boldsymbol{F}_b = -\nabla_{r_i}U_b$ and $\boldsymbol{F}_g = -\nabla_{r_i}U_g$ at the vertex $i$, respectively (Chirico & Langowski, 1994; Sano *et al.*, 2017).

We note that the increase of bending rigidity with time associated with lignification of the secondary cell wall discussed in Chelakkot and Mahadevan (2017) is not considered in this study because it has a minor effect on the stationary morphology. The experimentally observed temporal delay of gravitational sensing discussed in Agostinelli *et al.* (2020), Chauvet *et al.* (2019) is also not considered because our focus is not the detailed fitting of the bending process but the extraction of mechanics with qualitatively consistent bending behaviour.

To clarify how parameters in the model affect the shoot morphologies, we changed the parameters in the model. Through an extensive parameter search, we showed how the following six parameters affected the shoot morphology (Figure SI3). Note that though these parameters affect the shape morphology, the variety of shapes can be summarized by the two dimensionless parameters $S$ and $\varepsilon$. The first three paramters are growth-related parameters $l_0$, $l_g$ and $\dot{L}$, defined as the initial length, growing zone measured from tip and the linear growth rate, respectively (illustrated in Figure SI4). The other parameter $l_e$ is the elasto-gravity length arising from the competition between gravity and elasticity. The remaining two parameters, $\beta$ and $\gamma$, are curvature-related parameters defined as sensing strength of inclination angle and curvature, respectively.

## 4.4 The exhaustive search for the best-fitted parameters S and ε

To detect the best-fitted parameters S and $\varepsilon$ of the model $(S_{model}, \varepsilon_{model})$, we introduced a deviation $D$ between data and model with fixed $(S_{data}, \varepsilon_{data})$ as follows

$$D = \sqrt{\left(S_{\text{data}} - S_{\text{model}}\right)^2 + \left(\varepsilon_{\text{data}} - \varepsilon_{\text{model}}\right)^2}.$$

To search the parameter $\varepsilon_{\mathrm{model}}$, the mass density per unit length $\rho$ was systematically changed. To search the parameter $S_{\mathrm{model}}$, as the gravi-sensitivity $\beta$ and the proprioceptive sensitivity $\gamma$ are interrelated, we firstly searched both parameters $\beta$ and $\gamma$ and found a minimum deviation of $D$ around the range $S \in [2.0 \sim 4.0]$ for all examples of the wild type and the mutant (see all the detected values $(\beta_{\mathrm{model}}, \gamma_{\mathrm{model}})$ and detected $S$ in Figure SI7). As the parameter $S$ is expected to be constant, and we detected that the parameters $\beta$ and $\gamma$ have a relationship $\gamma_{\mathrm{model}} \sim 0.078\beta_{\mathrm{model}}$ for all the data, we fixed the average $S_{\mathrm{model}} \simeq 3.15 \left( = \frac{<l_g>}{0.078}, <\cdot> \text{ is sample average}\right)$ as an representative value and searched only the parameter $\beta$ for each individual.

## Acknowledgements

This work was supported by MEXT KAKENHI Grant-in-Aid for Scientific Research on Innovative Areas 'Plant Structure Optimization Strategy' Grant Numbers JP18H05484 and JP18H05488 (to MTM and TD). HS acknowledges financial support by MEXT KAKENHI JP19H05359 and by MEXT KAKENHI JP19K03766. TGS acknowledges financial support in the form of Grants-in-Aid for JSPS Overseas Research Fellowship (2019-60059).

**Financial Support.** This work was supported by MEXT KAKENHI JP18H 05484 and JP18H05488, by MEXT KAKENHI 19H05359 and by MEXT KAKENHI 19K03766.

**Conflicts of Interest.** The authors declare no conflicts of interest.

**Authorship Contributions.** Conception and design of the study, S.T., H.S., T.G.S., M.T.M and T.D.; Data quantification, S.T.; Construction of computational model, S.T., H.S. and T.G.S.; Writing of the manuscript, S.T., H.S., T.G.S., M.T.M. and T.D.

**Data Availability Statement.** The data that support the findings of this study are available from the corresponding author, Satoru Tsugawa, upon reasonable request. We share the simulation code that calculates the shoot morphologies on web browsers at github (https://satorutsugawa.github.io/linevertex.html).

**Supplementary Materials.** To view supplementary material for this article, please visit http://dx.doi.org/10.1017/qpb.2020.5.

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
