## [Reviewer Report]

Dear Editor:

Please find the enclosed manuscript titled “A data-validated mathematical model details the contributions of bending and stretching forces to gravitropic shoot bending in Arabidopsis” by Satoru Tsugawa, Tomohiko G Sano, Hiroyuki Shima, Miyo Terao Morita, and Taku Demura. We would like to submit the manuscript to Quantitative Plant Biology.

Plant shoot gravitropism is a complex phenomenon resulting from gravity sensing, curvature sensing (proprioception), and the ability to uphold self-weight. To better understand the mechanics of shoots during bending, we combined morphological data with a theoretical model. Using the wild type and lazy1-like 1 mutant Arabidopsis thaliana, we searched the best fitted theoretical parameters to relate the data to mathematical model. Based on the obtained models for the wild type and mutant, we discovered that both the bending force and the stretching force differ significantly between the wild type and mutant. We discuss the implications of the mechanical forces associated with differential cell growth and present a plausible mechanical explanation of shoot gravitropism.

We believe our article would fit nicely in Quantitative Plant Biology. In this article, we present a possible methodology to extract the mechanics of the shoot bending which is invisible in experiments. One of the original aspects of this study in comparison to previous works is the focus on the extraction of mechanical force during the actual shoot bending event. We found that both the bending force (derived from the gravi-proprioceptive response) and the stretching force (derived from shoot axial growth) are significantly different between the wild type and the lzy1 mutant. Moreover, it is revealed that the the differential cell growth underlying the bending event can be interpreted as a combined result of the gravi-proprioceptive response and the growth-induced bending, which correspond to the bending force and the stretching force, respectively. As a consequence of the two bending effects, we can understand the dynamic shoot morphology in terms of mechanics. These mechanical aspects of bending are the underlying mechanism hidden in the experimental description of shoot gravitropism.

We believe that this work is an important contribution to the environmental response in plant, the computational biology, and their interfacial communities especially in connecting the actual data and mathematical model, and should be of significant interest to the broad scientific readership in Quantitative Plant Biology.

We would be grateful if the manuscript can be reviewed in depth and considered for the publication in Quantitative Plant Biology.

Sincerely yours,

Satoru Tsugawa

Division of Biological Science,

Graduate School of Science and Technology, 

Nara Institute of Science and Technology,

Japan

---

## [Reviewer Report]

*Comments to Author*: This manuscript reports combined experimental and theoretical analysis to dissect mechanical forces acting on the shoot undergoing gravitropism. To understand the mechanics of gravitropism, morphodynamic analyses have been conducted before, yielding the differential growth dynamics explaining the observed series of deformations over the time course. However, the mechanical forces acting on the stem structure, especially the dynamic shift of them, remain to be elucidated. To do so the authors extended the mathematical model-based approach from Chelakkot and Mahadevan (2017), examining the wild type and the shoot gravitropism mutant lazy1. Based on the findings, they propose a mechanical mechanism of shoot gravitropism, which is a combination of bending force due to the gravitropism and proprioception and growth induced stretching force.

I find the method used in this work fascinating (different from mass-spring or finite-element analysis), and the findings are interesting and informative, especially the comparison between the wild type and the new mutant lazy1. And the model sound plausible. However, I would like to recommend further clarifications on some major and minor points outlined below.

Major points:

1. In the materials and methods, the mathematical model reads like it is basically the one developed by Chelakkot and Mahadevan (2017) (‘we review’ …). Would you clarify what part of the model and analysis was unique to your work?

2. I cannot find any reference to measured value of tissue stiffness or elasticity for wild type or lazy stem, or the weight of the stem, and so I assumed that the forces indicated here are relative, based on the assumption of uniform and equal elasticity for both wild type and lazy1. However, I do see N.min in Fig 3E-G. How is it possible? Can you explain more clearly how you get to the absolute value?

3. I consider the analysis of lazy1 is a major strength of this work, but I find the wild type-mutant comparison is not effectively utilised in this manuscript. Often the text says ‘it was different in wild type compared to the mutant’ – in what way is it different? From this work, what can you say about the defects in lazy1; what specific mechanical processes is it impaired? For example, Fig 4D seems to be showing an interesting difference between the two genotypes, illuminating what lazy1 cannot do in the course of the complex process called gravitropism.

Minor points:

1. Some labels in the figures are too small and hard to see without digital magnification.

2. I don’t quite understand why you say that the shift between the growth rates of the inner and outer flank (the final paragraph, P9) is surprising – isn’t it expected if the stem bending first overshoot and then de-bend?

3. Please do consider sharing your data and code. This (brand new!) journal prides in its open access, and also the amount of data and code are not so large to challenge sharing on public database.

---

## [Reviewer Report]

*Comments to Author*: This work aims to understand the mechanics underlying gravitropism, by combining morphological data with a theoretical model to analyze shoot bending in wild-type and lazy1-like 1 mutant Arabidopsis thaliana. Using data from actual bending events, the authors searched for model parameters that minimized discrepancies between the data and mathematical model. They find that both the bending force (derived from the gravi-proprioceptive response) and the stretching force (derived from shoot axial growth) differ significantly between the wild type and mutant.

I have a number of issues with the analysis done in this manuscript, but perhaps I have misunderstood and I hope the authors will be able to clarify:

Major comments:

1. I am concerned about the statistics: 3 experiments for wild type and mutant, is vary scarce in terms of data. You would need at least 3-4 times more in order to make any statistically significant conclusions.

2. Line 151: It says that “we systematically searched through values for the dimensionless parameters and to find those that resulted in the minimum deviation between the model and data. “

The model they use by Chelakkot et al, is based on the model by Bastien et al, where two parameters describe the kinematics of gravitropism: gravi-sensitivity beta, and proprioceptive sensitivity gamma. These must be known in order to use Chelakkot’s model. What are these values? Do they fit them or assume them?

Since lazy1 mutant lacks the ability to sense gravity, this should significantly affect beta, however would imaging that the mechanical responses (which should be passive in nature) should not change dramatically.

3. Line 160: “The model predicts the value ranges of the curvature (Figs 2E and 2F), which are consistent with the real data (Fig. 1C and 1D). “

How is it consistent? how was the model prediction compared to the data?

4. Line 165: it says that the total displacement of the shoot is due to stretching forces, bending forces, and gravity.

Here I am deeply confused since the Chelakkot model explicitly assumes an inextensible and unshearable thin filament, where only an external load of gravity acts on the shoot. This means that it is assumed there are no internal stresses. Indeed just before eq 2.8 in Chelakkot et al it says “The filament is kept practically inextensible by setting E ≫ B.”

Therefore I am not sure how this model can be used for stretching forces…

5. Line 191: “bending and stretching forces can be translated into biological counterparts.”

I find this problematic. The model by Chelakkot et al assumes that there are active biological processes (gravisensing and proprioception), and *passive* elastic responses due to gravity. But the two are not related. I feel there is a mix between mechanical forces and biological counterparts. Of the course the two may be connected (i.e. proprioception may be due to sensing of internal stresses), but since the current work does not discuss how grave-sensitivity beta was found, and introduced stretching in a model which is valid for non stretching filaments… I find the conclusion problematic.

6. Line 208: “As the shoot grows, the current curvature becomes different from the intrinsic curvature ∗ “

I think this statement is wrong: my understanding is that as the shoot grows it changes the intrinsic curvature ∗ (through a material derivative - where material flows, there is no bending), the actual curvature kappa is governed by passive elasticity (as discussed before).

Minor comments:

Line 61: “data-analytical” and “model-analytical” approaches - I find these are not good terms… perhaps “data-driven” and “model-driven” approaches?

Line 82-84: Important to stress that the model by Bastien et all considers only active processes, while the model by Chellakot et al is based on this model, adding effects of elasticity.

Line 94: “Next, we combined the data with the model to extract mechanical forces…”

What model? This comes out of the blue and is unclear to the reader.

Line 112: a morphospace should be explained a bit better - most readers will not be familiar with the concept.

In all I find this work problematic both from the statistics point of view, and particularly in terms of the model and how it was implemented. Unfortunately I do not feel it can be accepted in its current form, unless significant clarifications are made.

---

## [Reviewer Report]

*Comments to Author*: The reviewers made a number of points regarding the clarity of the description of your model and the presentation of your results. There is also a concern with the statistical significance of your experimental results. These points should be addressed in your revision.

---

## [Reviewer Report]

Dear Editor:

Please find the enclosed manuscript titled “A data-validated mathematical model details the contributions of bending and stretching forces to gravitropic shoot bending in Arabidopsis” by Satoru Tsugawa, Tomohiko G Sano, Hiroyuki Shima, Miyo Terao Morita, and Taku Demura. We would like to submit the manuscript to Quantitative Plant Biology.

Plant shoot gravitropism is a complex phenomenon resulting from gravity sensing, curvature sensing (proprioception), and the ability to uphold self-weight. To better understand the mechanics of shoots during bending, we combined morphological data with a theoretical model. Using the wild type and lazy1-like 1 mutant Arabidopsis thaliana, we searched the best fitted theoretical parameters to relate the data to mathematical model. Based on the obtained models for the wild type and mutant, we discovered that both the bending force and the stretching force differ significantly between the wild type and mutant. We discuss the implications of the mechanical forces associated with differential cell growth and present a plausible mechanical explanation of shoot gravitropism.

We believe our article would fit nicely in Quantitative Plant Biology. In this article, we present a possible methodology to extract the mechanics of the shoot bending which is invisible in experiments. One of the original aspects of this study in comparison to previous works is the focus on the extraction of mechanical force during the actual shoot bending event. We found that both the bending force (derived from the gravi-proprioceptive response) and the stretching force (derived from shoot axial growth) are significantly different between the wild type and the lzy1 mutant. Moreover, it is revealed that the the differential cell growth underlying the bending event can be interpreted as a combined result of the gravi-proprioceptive response and the growth-induced bending, which correspond to the bending force and the stretching force, respectively. As a consequence of the two bending effects, we can understand the dynamic shoot morphology in terms of mechanics. These mechanical aspects of bending are the underlying mechanism hidden in the experimental description of shoot gravitropism.

We believe that this work is an important contribution to the environmental response in plant, the computational biology, and their interfacial communities especially in connecting the actual data and mathematical model, and should be of significant interest to the broad scientific readership in Quantitative Plant Biology.

We would be grateful if the manuscript can be reviewed in depth and considered for the publication in Quantitative Plant Biology.

Sincerely yours,

Satoru Tsugawa

Division of Biological Science,

Graduate School of Science and Technology, 

Nara Institute of Science and Technology,

Japan

---

## [Reviewer Report]

*Comments to Author*: The authors addressed my suggestions and comments mostly, and the revised descriptions provide key information that helps the readers understand their work.

I would like to suggest further changes (all minor, but helpful) before acceptance of this manuscript.

1) Response to Comment #1 -

“We note that the mathematical model itself in this study is exactly the same as the model in Ref. [24] but the methodology to fit the model to spatio-temporal bending data described below is novel in the field of gravitropism.”

< Do you mean '... but fitting experimentally obtained bending dynamics data to this model is novel in the context of gravitropism'?

2) Response to Comment #2 -

'we used a relative value E=40B with a normalized value B=1 (N⋅m)'

< Why did you use this particular value? Please justify. Is it based on measurement with the Arabidopsis stem with comparable stages of development as observed in the bending dynamics study? And do you assume the rigidity is the same for lazy1 mutant (which I would not expect) - if it is OK to assume so, please explain your rationale.

3) Response to Comment #3 -

'To clarify the term “difference”, we revised the corresponding statement using “weaker” in terms of maximum stretching force and maximum bending force at line 206, and subsequently added a mention to the defects in lzy1 mutant as below.

“Accordingly, the maximum stretching force and maximum bending force are significantly weaker in lzy1 mutant compared to those in the wild type (Fig. 3E, 3F, 3G). That means that the defects in the signal transduction from the gravi-sensing in lzy1 mutant is reflected in the weaker magnitudes of stretching and bending force.”'

< 'Weaker' is more specific than 'different', but it can be made even more specific. lzy1 is not devoid of the gravitropic response, but to what extent (e.g. %), and affected at what stages and parts of the organ more (spatio-temporal specificity of the phenotype)?

---

## [Reviewer Report]

*Comments to Author*: Unfortunately I did not find the response of the authors satisfactory - in fact I am more concerned than before regarding a few points.

The main issues are:

- the description of the analysis and comparison between data and model (the heart of this paper) is qualitative only - there is no quantitive analysis available.

- the use of ‘forces’ and the terms ‘bending’ and ‘stretching’ in order to describe the active growth-driven processes of elongation and tropic bending is confusing and inaccurate. These are not considered forces in the models on which the analysis is based (Bastien et al, Chelakkot et al).

Furthermore please note that the supplementary figures were not included.

Below I bring more detailed comments to some of their responses:

Comment #2

The process is still not clear. For example when you say that you find the “minimum deviation” - what exactly do you mean? I would expect to see the definition of this deviation, and a plot of it vs. values of S with a clear minimum value around S~3.15. I get the feeling this is a qualitative analysis only.

Another unrelated point for concern is the fact that the error in the computed beta is such that it does not allow to discern between the value found for wild type and lzy1.

Comment #3: Similar to the previous comment. You say that “we explicitly wrote ether value range and patio-temporal tried in curvature” - from this I understand that the comparison is qualitative only. In this case it is also no surprising that the model is consistent with the data, since it was constructed to reproduce these general qualitative features.

Comment #4, #5

It still seems like there is some confusion with the terms.

The change in curvature due to active processes (gravitropism) is not due to bending forces - rather it is due to differential growth, where one side of the shoot grows at a higher rate than the other. A common assumption, especially in the model by Bastien et al, is that there are no internal stresses etc.

Therefore the whole use of forces, and the terms bending and stretching, instead of terms like curvature and length, is extremely confusing and inaccurate.

---

## [Reviewer Report]

*Comments to Author*: Dear Dr. Tsugawa,

Your revised manuscript "A data-validated mathematical model details the contributions of bending and stretching forces to gravitropic shoot bending in Arabidopsis" has been fully reviewed by the two initial reviewers.

While reviewer #1 is mostly satisfied with your revised version and recommends only minor revisions, reviewer #2 has stronger objections. In addition to all suggestions made by both reviewers, I believe two specific aspects should be substantially improved:

1) Can you perform a quantitative analysis to model predictions to data? If this is not possible or too demanding, could you reframe your manuscript as a "Theory" paper and propose a protocol for a quantitative test of your model (for example as a new subsection in "Results")?

2) Can you clarify how you handle mechanics and how it relates to the models on which you based your work (Bastien et al. and Chelakkot et al.)? In these two models, growth (and especially differential growth) is not described in terms of forces, but in terms of pure kinematics (curvature and length). If you have followed a different approach than Bastien et al. and Chelakkot et al. to model growth (with bending and stretching), then you should explain how your model deviates from the ones your are referring to.

Sincerely,

Félix Hartmann

---

## [Reviewer Report]

Dear Editor:

Please find the enclosed manuscript titled “A mathematical model details the contributions of bending and stretching forces to gravitropic shoot bending in Arabidopsis” by Satoru Tsugawa, Tomohiko G Sano, Hiroyuki Shima, Miyo Terao Morita, and Taku Demura. We would like to submit the manuscript to Quantitative Plant Biology.

Plant shoot gravitropism is a complex phenomenon resulting from gravity sensing, curvature sensing (proprioception), and the ability to uphold self-weight. To better understand the mechanics of shoots during bending, we combined morphological data with a theoretical model. Using the wild type and lazy1-like 1 mutant Arabidopsis thaliana, we searched the best fitted theoretical parameters to relate the data to mathematical model. Based on the obtained models for the wild type and mutant, we discovered that both the bending force and the stretching force differ significantly between the wild type and mutant. We discuss the implications of the mechanical forces associated with differential cell growth and present a plausible mechanical explanation of shoot gravitropism.

We believe our article would fit nicely in Quantitative Plant Biology. In this article, we present a possible methodology to extract the mechanics of the shoot bending which is invisible in experiments. One of the original aspects of this study in comparison to previous works is the focus on the extraction of mechanical force during the actual shoot bending event. We found that both the bending force (derived from the gravi-proprioceptive response) and the stretching force (derived from shoot axial growth) are significantly different between the wild type and the lzy1 mutant. Moreover, it is revealed that the the differential cell growth underlying the bending event can be interpreted as a combined result of the gravi-proprioceptive response and the growth-induced bending, which correspond to the bending force and the stretching force, respectively. As a consequence of the two bending effects, we can understand the dynamic shoot morphology in terms of mechanics. These mechanical aspects of bending are the underlying mechanism hidden in the experimental description of shoot gravitropism.

We believe that this work is an important contribution to the environmental response in plant, the computational biology, and their interfacial communities especially in connecting the actual data and mathematical model, and should be of significant interest to the broad scientific readership in Quantitative Plant Biology.

We would be grateful if the manuscript can be reviewed in depth and considered for the publication in Quantitative Plant Biology.

Sincerely yours,

Satoru Tsugawa

Division of Biological Science,

Graduate School of Science and Technology, 

Nara Institute of Science and Technology,

Japan

---

## [Reviewer Report]

*Comments to Author*: Dear Dr. Tsugawa,

After a close examination of your revisions and responses to the reviewers, I believe that your new manuscript can be accepted with minor revisions.

You could be more modest in your statements. For instance, the manuscript title ("A mathematical model details the contributions of bending and 2 stretching forces to gravitropic shoot bending in Arabidopsis") could be amended into "A mathematical model explores the contributions of bending and stretching forces to gravitropic shoot bending in Arabidopsis".

In the Abstract, the sentence "Based on the resulting model, we discovered that both the bending force (derived from the gravi-proprioceptive response) and the stretching force (derived from shoot axial growth) differ significantly between the wild type and mutant." could be changed into "The resulting model suggests that both the bending force (derived from the gravi-proprioceptive response) and the stretching force (derived from shoot axial growth) differ significantly between the wild type and mutant."

You could improve the quantitative section "Future perspective: prediction for growth at the cellular level". Please explain in which way your quantitative output (the differential growth) is a non-obvious prediction of your model: Not a measurement from image analysis, but an experimentally testable result of your mechanical modeling.

Sincerely,

Félix Hartmann

---

## [Reviewer Report]

Dear Editor:

Please find the enclosed manuscript titled “A mathematical model details the contributions of bending and stretching forces to gravitropic shoot bending in Arabidopsis” by Satoru Tsugawa, Tomohiko G Sano, Hiroyuki Shima, Miyo Terao Morita, and Taku Demura. We would like to submit the manuscript to Quantitative Plant Biology.

Plant shoot gravitropism is a complex phenomenon resulting from gravity sensing, curvature sensing (proprioception), and the ability to uphold self-weight. To better understand the mechanics of shoots during bending, we combined morphological data with a theoretical model. Using the wild type and lazy1-like 1 mutant Arabidopsis thaliana, we searched the best fitted theoretical parameters to relate the data to mathematical model. Based on the obtained models for the wild type and mutant, we discovered that both the bending force and the stretching force differ significantly between the wild type and mutant. We discuss the implications of the mechanical forces associated with differential cell growth and present a plausible mechanical explanation of shoot gravitropism.

We believe our article would fit nicely in Quantitative Plant Biology. In this article, we present a possible methodology to extract the mechanics of the shoot bending which is invisible in experiments. One of the original aspects of this study in comparison to previous works is the focus on the extraction of mechanical force during the actual shoot bending event. We found that both the bending force (derived from the gravi-proprioceptive response) and the stretching force (derived from shoot axial growth) are significantly different between the wild type and the lzy1 mutant. Moreover, it is revealed that the the differential cell growth underlying the bending event can be interpreted as a combined result of the gravi-proprioceptive response and the growth-induced bending, which correspond to the bending force and the stretching force, respectively. As a consequence of the two bending effects, we can understand the dynamic shoot morphology in terms of mechanics. These mechanical aspects of bending are the underlying mechanism hidden in the experimental description of shoot gravitropism.

We believe that this work is an important contribution to the environmental response in plant, the computational biology, and their interfacial communities especially in connecting the actual data and mathematical model, and should be of significant interest to the broad scientific readership in Quantitative Plant Biology.

We would be grateful if the manuscript can be reviewed in depth and considered for the publication in Quantitative Plant Biology.

Sincerely yours,

Satoru Tsugawa

Division of Biological Science,

Graduate School of Science and Technology, 

Nara Institute of Science and Technology,

Japan